

# Stress distribution in the bonobo (*Pan paniscus*) trapeziometacarpal joint during grasping

Timo van Leeuwen[1,2], G. Harry van Lenthe[2], Evie E. Vereecke[1] and Marco T. Schneider[3]

[1] Department of Development and Regeneration, KU Leuven, KULAK, Kortrijk, Belgium
[2] Department of Mechanical Engineering: Biomechanics Section, KU Leuven, Leuven, Belgium
[3] Auckland Bioengineering Institute, University of Auckland, Auckland, New Zealand

Corresponding author
Timo van Leeuwen,
timo.vanleeuwen@kuleuven.be

## ABSTRACT

The primate thumb plays a central role in grasping and the basal trapeziometacarpal (TMC) joint is critical to its function. The TMC joint morphology varies across primates, yet little is known about form-function interaction within in the TMC joint. The purpose of this study was to investigate how stress distributions within the joint differ between five grasping types commonly employed by bonobos (*Pan paniscus*). Five cadaveric bonobo forearms were CT scanned in five standardized positions of the hand as a basis for the generation of parametric finite element models to compare grasps. We have developed a finite element analysis (FEA) approach to investigate stress distribution patterns in the TMC joint associated with each grasp type. We hypothesized that the simulated stress distributions for each position would correspond with the patterns expected from a saddle-shaped joint. However, we also expected differences in stress patterns arising from instraspecific variations in morphology. The models showed a high agreement between simulated and expected stress patterns for each of the five grasps (86% of successful simulations), while partially (52%) and fully (14%) diverging patterns were also encountered. We identified individual variations of key morphological features in the bonobo TMC joint that account for the diverging stress patterns and emphasized the effect of interindividual morphological variation on joint functioning. This study gives unprecedented insight in the form-function interactions in the TMC joint of the bonobo and provides an innovative FEA approach to modelling intra-articular stress distributions, a valuable tool for the study of the primate thumb biomechanics.

## INTRODUCTION

The combination of high thumb mobility and forceful grasping is considered a unique modern human hallmark (*Marzke, 1997*; *Napier, 1956*; *Napier, 1960*; *Napier & Napier, 1967*; *Susman, 1998*). The high mobility of the modern human thumb predominantly originates from the morphology of the basal thumb *i.e.* the trapeziometacarpal (TMC) joint (Fig. 1) (*Napier, 1955*). The TMC joint is a reciprocal saddle-shaped joint, the

physiological equivalent of a universal joint which allows motion in two planes (flexion-extension, abduction-adduction) (*Cooney & Chao, 1977*). The shape allows for large ranges of motion of the first metacarpal (MC1), but also offers the stability required for forceful grasping (*D'Agostino et al., 2017*; *Napier, 1955*; *Rose, 1992*). The morphology of the TMC joint varies across primates (*Lewis, 1977*; *Marzke et al., 2010*; *Rafferty, 1990*), but how this relates to specific thumb function in these primates remains unclear. Where many studies focus on the human TMC joint, especially from a clinical point of view (*Crisco et al., 2015*; *Dourthe et al., 2016*; *Haines, 1944*; *Kerkhof et al., 2018*; *Kuczynski, 1974*; *Ladd et al., 2013*; *Ladd et al., 2014*; *Schneider et al , 2015*), far less is known about form-function relations in the non-human primate thumb. In this study, we investigate the effect of different grip types on the stress distribution in the TMC joint in bonobos (*Pan paniscus*, fam. *Hominidea*) (Fig. 2A). Bonobos are both arboreal and terrestrial species that use their hands in both locomotion and manipulation, including cases of tool use that have been observed in captivity. (*Bardo et al., 2016*; *Christel, Kitzel & Niemitz, 1998*; *Crast et al., 2009*; *Feix et al., 2015*; *Ingmanson, 1998*; *Jordan, 1982*; *Neufuss et al., 2017*). Common grasp types used by bonobos include various types of power grasps as well as a key pinch precision grip (Fig. 2) (*Christel, 1993*; *Christel, Kitzel & Niemitz, 1998*; *Napier, 1956*; *Neufuss et al., 2017*). In addition, bonobos use their thumb in vertical climbing and clambering, to grasp trunks or branches, but the thumb is not loaded during knuckle-walking (*Inouye, 1992*; *Samuel et al., 2018*; *Tuttle, 1967*; *Wunderlich & Jungers, 2009*).

Bonobos, like humans, possess a saddle-shaped TMC joint, and have a similar soft-tissue configuration, *i.e.,* the surrounding muscles and ligaments, to that of humans (*van Leeuwen et al., 2018*; *van Leeuwen et al., 2019*). Yet, such morphology is not unique to hominids. In fact the saddle-shape is also present in New World monkeys (*Lewis, 1977*; *Rafferty, 1990*), some of which are also highly dextrous (*Parker & Gibson, 1977*). In bonobos, the TMC joint is highly curved even when compared to humans (*Marzke, 1997*), allowing for wide ranges of motion in particular during adduction and adduction of the thumb. Our previous research on the kinematics of the primate TMC joint has shown that the mobility of the bonobo thumb is similar to that of humans, but thumb extension is limited by the presence of a strong anterior oblique ligament which reinforces the palmar side of the TMC joint (*van Leeuwen et al., 2019*). Based on these findings, we expect that the force distribution within the TMC joint will reflect signals of the specific joint morphology. Recently, *Schneider et al. (2017)* developed a finite element (FE) approach to model stress distribution in the human TMC joint. Here, we build upon this work and propose a derived model-based method to investigate the stress distribution in the bonobo TMC joint.

In total, we investigate five types of grasps commonly used by bonobos (*Christel, 1993*; *Samuel et al., 2018*). Out of these, three are large diameter power grasps, one with the thumb abducted, one with the thumb extended, and one with an adducted thumb. The abducted type, further referred to as *powerL* (Fig. 2B), is characterized by a wide grip and an abducted and externally rotated thumb that is still radioulnarly in line with the trapezium. Stresses are expected to be diverted to the radio-central aspect of the articular facet of the trapezium during *powerL*. The large diameter power grasp with extended thumb (*powerLE*) (Fig. 2C) extends the thumb near the maximal excursion of the TMC joint, but in line
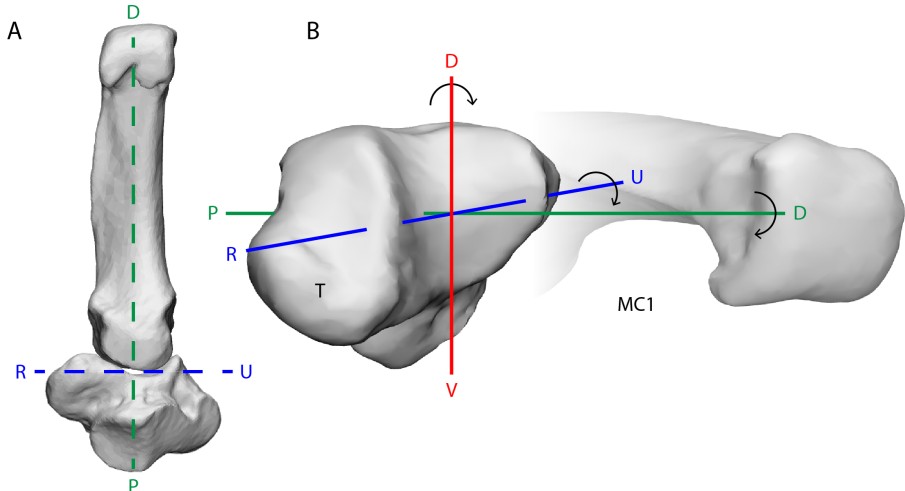

**Figure 1  The bonobo trapeziometacarpal (TMC) joint from a ventral (palmar) view (A) and a radiodistal view (B).** Its reciprocal surfaces are saddle-shaped. The distal articular surface of the trapezium (T) has a convex curvature along the dorsal-palmar (D–V) axis, and makes contact with the concave curvature of the first metacarpal (MC1), while the trapezial articular surface along the radio-ulnar (R–U) axis is concave, to match the convex curvature of the MC1. The principal movements of the thumb align with rotation about one of three axes of the joint; rotation about the D–V axis results in palmar abduction-adduction, rotation about the R–U axis results in thumb flexion-extension, and rotation about the proximal-distal (P–D) axis results in axial rotation of the thumb. The depicted bone models are trapezium and MC1 surface meshes reconstructed from CT-scanning Pp5 in a neutral position of the thumb.

with the dorsopalmar plane of the trapezium. Stress peaks are therefore expected on the dorsal-central aspect of the articular facet of the trapezium. The large diameter power grasp with thumb adducted (*powerLA*) (Fig. 2D) places the thumb in line with the other digits of the hand. The thumb is adducted and externally rotated. Here, the opposite is expected when compared to grasps with an abducted thumb; high stress in the ulnar-central articular facet of the trapezium. The small diameter power grasp (*powerS*) (Fig. 2E) is similar to the large diameter variant, but in contrast, the amount of thumb abduction is less extreme. Stress is expected to concentrate centrally on the articular facet, with a radial tendency. The key *pinch* (Fig. 2F) is characterized by a relatively straight thumb and trapeziometacarpal joint. External thumb tip force is exerted downwards towards the radial side of the middle phalanx of a flexed index finger. As a result, forces are expected to be diverted towards to the central-palmar aspect of the trapezial facet, creating stress peaks in this area. A visual representation of the expected stress patterns is included in the 'Results' section.

We simulate the stress distribution within the TMC joint during these five positions of the hand in five cadaveric bonobo forearms. We hypothesize that the stress distribution within the TMC joint for each individual grasp reflects the expected distribution for a universal joint during that grasp, as described above. Furthermore, we expect each individual stress distribution pattern to be influenced by the specimen's individual joint morphology. Therefore, we hypothesize that deviations from the predicted stress patterns for each grasp can be related to differences in TMC joint morphology.
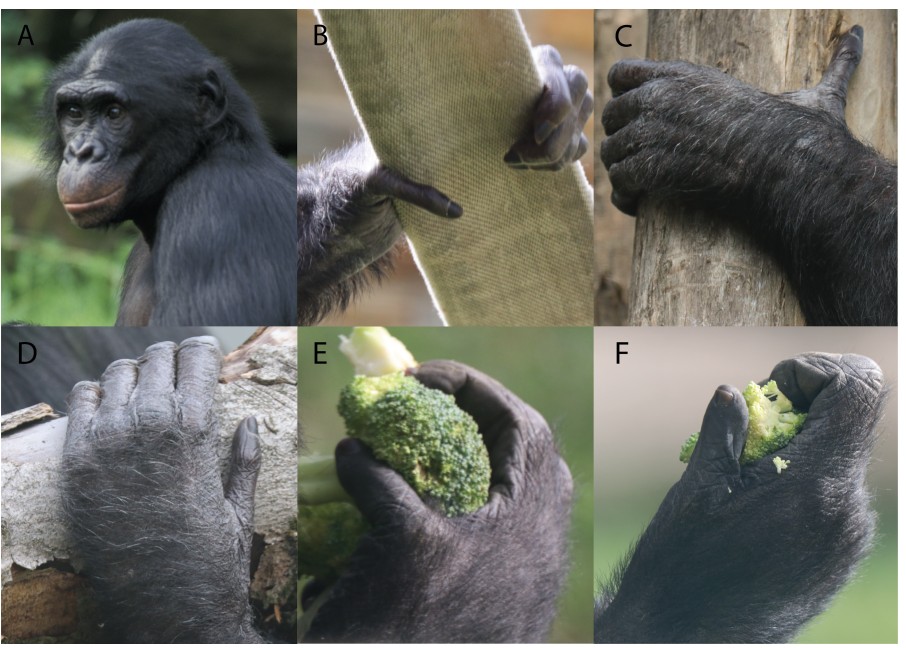

**Figure 2** **Depiction of the different grasping types observed in bonobos.** (A) An adult male bonobo, (B) large diameter power grip with thumb abducted (*powerL*), (C) large diameter power grip with thumb extended (*powerLE*), (D) large diameter power grip with thumb abducted (*powerLA*), (E) small diameter power grip (*powerS*), (F) pad-to-side precision grip or keypinch (*pinch*). Photography by courtesy of Marie Vanhoof and prof. Kristiaan D'Aout.

# MATERIALS & METHODS

In this study, we use an FE approach to model TMC joint stress distributions during various types of grasps employed by bonobos in manipulation and arboreal locomotion. The focus of this study is on the distribution of stress within the TMC joint during each grasp, the localisation of stress peaks, and how stress distributions may be influenced by individual TMC joint morphology. We use an optimised parametric FE approach to model TMC joints of five cadaveric bonobo hands in five functional positions.

## Specimens

Five cadaveric forearms of (sub)adult bonobos were obtained opportunistically *via* collaboration with five European zoos (Table 1). The forearms were disarticulated either at the glenohumeral joint or at the elbow joint and stored at −18 °C shortly after decease. All specimens were part of parallel studies on bonobo forelimb morphology and thumb biomechanics (*van Leeuwen et al., 2018*; *van Leeuwen et al., 2019*). Specimen Pp2 was a sub-adult with a TMC joint that was not fully mature, presenting incomplete epiphysial fusion of the MC1 and a relatively smooth articular periphery (without pronounced bony ridges). The specimen has been included in the sample as the articular surfaces were sufficiently developed to present the reciprocal saddle-shape of the adult TMC joint. No articular pathologies were observed at the TMC joint in any of the specimens used for this study.
**Table 1 Specimen information.**

| Code | Subject ID | Age | Sex | Sample | Origin | Zoo |
|------|-----------|-----|-----|--------|--------|-----|
| Pp1 | 8365526 | 35 | M | L | Wild born[***] | Wilhelma Zoo, Stuttgart, Germany |
| Pp2 | 15295295 SB: 295 | 8 | F | L | Captive born | Royal Zoological Society Antwerp, Belgium |
| Pp3 | MIG12-29882197 SB: 126 | 32 | M | R | Captive born | Zoo Frankfurt, Germany |
| Pp4[*] | SB:177 CITES:79209 | 24 | M | R | Captive born | La Vallée des Singes, Le Gureau, France |
| Pp5[**] | SB: 88 27641621 | 39 | F | L | Wild born[***] | Wilhelma Zoo, Stuttgart, Germany |

**Notes.**
[*]Master specimen
[**]Scanned in 4 positions only
[***]Wild born but raised in captivity
SB, studbook number; M, male; F, female; R, right; L, left.

## Imaging protocol

Each specimen was thawed at room temperature over a period of 24 h before being imaged using a 64-slice Discovery HD 750 CT scanner (GE Healthcare, Little Chalfont, UK; display field of view (DFOV): 173 mm; slice thickness: 0.625 mm; pixel spacing: 0.338/0.338 mm, voxel size: 0.071 mm$^3$; 100 kV; 180 mA; image size: 512 × 512 pixels). A radio-translucent rig with a small (50 mm) and large diameter peg (125 m) was used to standardize each hand in the five functional grip positions (*i.e., powerL, powerLA, powerLE, powerS,* and *pinch*). This resulted in a total of 24 scan positions, as we were unable to position specimen Pp5 in a *powerL* grasp. Each trapezium and the first metacarpal was segmented separately using Mimics Research 20.0 (Materialise, Leuven, Belgium) and exported as triangular meshes.

## Parametric meshing

Parametric FE meshes for 24 TMC joint models were generated in two steps by aligning all specimen-position combinations to a template set of TMC meshes and fitting the discretised master meshes to the 23 remaining specimen-position combinations.

### Step 1: Mesh alignment

For each specimen, the trapezium and MC1 STL meshes of one specific grasp (*pinch*) were selected and designated as template meshes (Fig. 3A). A master TMC set template mesh (Fig. 3A) was selected and reoriented to a TMC joint-based coordinate system where the principal directions of motion of the thumb (*i.e.,* abduction-adduction, extension-flexion, and axial rotation) are aligned with the cartesian coordinate system using a method based on *Halilaj et al. (2013)* and *van Leeuwen et al. (2019)*. The individual template sets for the remaining specimens were aligned with the master mesh in the same coordinate system by rigid body registration fitting the trapezium to the master trapezium template, and transforming the corresponding MC1 (Fig. 3B). The MC1s of all remaining specimen-position combinations were aligned with their respective template trapezium by rigid body registration fitting each combination's respective trapezium and transforming the MC1 using the resulting rotation matrix. This yields five aligned triangular trapezium meshes,
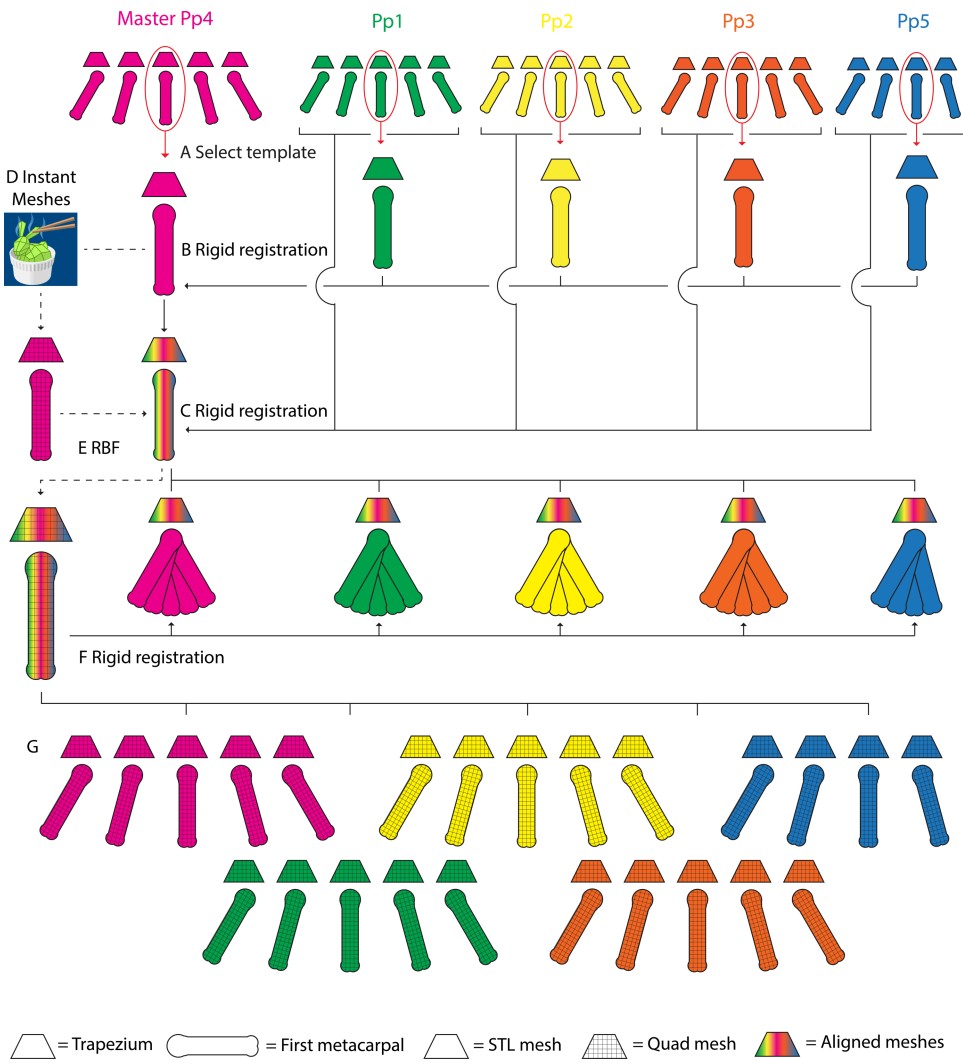

**Figure 3** **A schematic representation of the parametric finite element model generation pipeline.** Template TMC meshes are selected for each specimen, one set is designated as the master mesh (A). All template meshes are aligned with the master mesh using rigid body registration of the trapezia, and transforming the MC1s onto the master meshes (B). The remaining MC1 positions for each specimen are aligned with their corresponding template set, again using rigid body registration of the trapezia and transforming the MC1s (C). The master template mesh is discretized into quadrilateral meshes (D), the articular surfaces are identified, extracted as quadrilateral meshes, and lofted into hexahedral cartilage meshes (not visualized). The quadrilateral bone and hexahedral cartilage (not visualized) master templates are fitted to the morphology of each specimen's template set using Radial Basis Function (RBF) fitting (E). The master MC1 template meshes are fitted to the remaining grasping positions in each specimen using rigid body registration (F). The workflow yields discretized quadrilateral meshes for all five specimens, for each of the five (four in the case of Pp5) positions (G).

one per specimen, each with five corresponding triangular MC1 meshes, one per grasp (with the exception of Pp5, which has four) (Fig. 3C).

### Step 2: Mesh fitting

The trapezium and MC1 master template triangular meshes were discretized into quadrilateral meshes using Instant Meshes software (*Jakob et al., 2015*) (Fig. 3D). The reciprocal subchondral articular surfaces of the TMC joint were identified on both bones from bone landmarks, and delineated *via* local meshing using Instant Meshes' Position Field and Orientation Field tools. To ensure the subchondral surface mesh quality, Instant Meshes' Singularity tools were used to reduce mesh irregularities, minimize element distortion, and increase mesh Jacobian values (*Burkhart, Andrews & Dunning, 2013*). The delineated subchondral elements were extracted using MeshLab software (v2016.12) (*Cignoni et al., 2008*) and exported as separate quadrilateral meshes. Custom Python (3.6) scripts were used to generate quadrilateral template TMC meshes for each individual specimen. The discretised master trapezium, MC1, and their respective cartilage meshes were fitted to the vertices of each specimen's triangular template mesh using Radial Basis Function (RBF) fitting (Fig. 3E). A three-layer, hexahedral cartilage layer mesh was generated for each specimen by lofting the extracted subchondral meshes along the surface node normal using a custom Python script based on *Schneider et al. (2017)*. As cartilage information for the bonobo TMC joint was lacking, the cartilage layer was generated at a thickness of 0.5 mm, somewhat lower than the 0.6–0.7 mm thickness reported for human TMC joint cartilage (*Dourthe et al., 2019*) to account for the relatively smaller bone size. MC1 quadrilateral meshes for the remaining grasps of each specimen were generated by rigid body registration fitting their quadrilateral MC1 template mesh to each position (Fig. 3F), and using the resulting transformation matrices to transform the corresponding hexahedral cartilage meshes. This yields five quadrilateral trapezium meshes, one per specimen, each with five (with the exception of Pp5) corresponding quadrilateral MC1 meshes (Fig. 3G), one per position, and hexahedral cartilage meshes for all bone meshes.

## FE model generation

All FE models were generated and solved using FEBio Software Suite (*Maas et al., 2012*). A master model (Fig. 4) was manually set up in FEBio Preview (v2.1.3) to be solved by the FEBio solver (v.2.9.0). The quadrilateral bone surface meshes were modelled as rigid bodies, and connected to their respective hexahedral cartilage meshes, modelled as a neo-Hookean material (Young's modulus = 18 MPa, Poisson's ratio = 0.45) (*Black & Hastings, 2013*). The trapezium mesh was constrained in all degrees of freedom, while the MC1 was given three degrees of freedom, allowing it to translate but not rotate.

As the imaging was performed on cadaveric material, the resulting bone meshes were situated in unloaded positions. To account for this, the model was set up as a two-step, force-driven model. The first step serves to bring the elements into contact, facilitated by nonlinear springs, exerting 1N of force each, corresponding to the bonobo TMC joint ligaments (*van Leeuwen et al., 2019*). Step two introduces the applied force vector. We performed a 2D static force analysis, based on *Cooney & Chao (1977)*, to estimate the

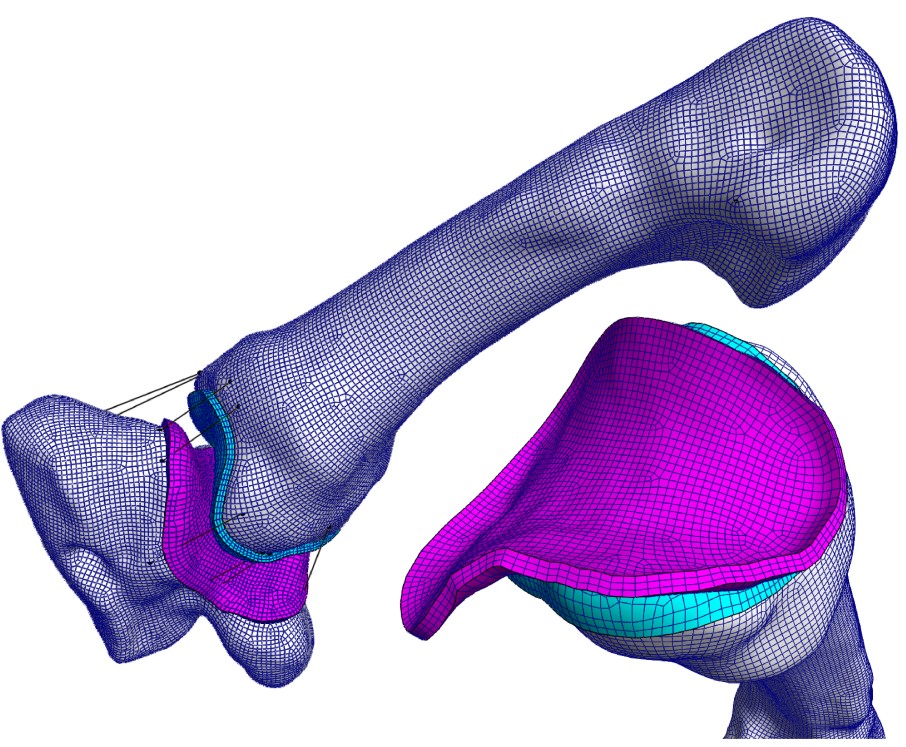

**Figure 4   The TMC joint FE model in the large diameter power grasp with extended thumb (*powerLE*).** The coloured surfaces represent the cartilage layers on the trapezium (pink) and MC1 (blue), black lines indicate non-linear springs and correspond to ligament positions. Shown here is the FE model based on the morphology of Pp4.

magnitude of the resultant force vector applied on the MC1. In the static force analysis, the initial thumb tip force was estimated at 50 N, based on the physiological cross-sectional area of the thenar muscles, and muscle attachment information was based on detailed dissections published by our research group (*van Leeuwen et al., 2018*). The analysis yielded a force vector magnitude of about 300 N, which was applied to the MC1 for all positions. Successful model termination was achieved when the FEBio solver was capable of executing all steps of the force application process without any elements collapsing.

The remaining 23 models were generated using a custom Python script that substitutes the master mesh data with the parametric mesh data and adapted the force vector orientation to conform to the first metacarpal orientation for each specimen-position combination.

To simulate the five grasping positions of the thumb, the models were optimized to bring the position after loading in close agreement with the scanned positions of the thumb. This was done using a Python-based function that optimizes the orientation of the force vector to yield a loaded terminating position that closely resembles the unloaded, scanned position of the MC1 by reducing the root mean square (RMS) error between these positions. As RMSE gives a relatively high weight to larger errors, it helps guide the optimiser towards the optimal position of the MC1. The proximo-distal difference between the loaded and

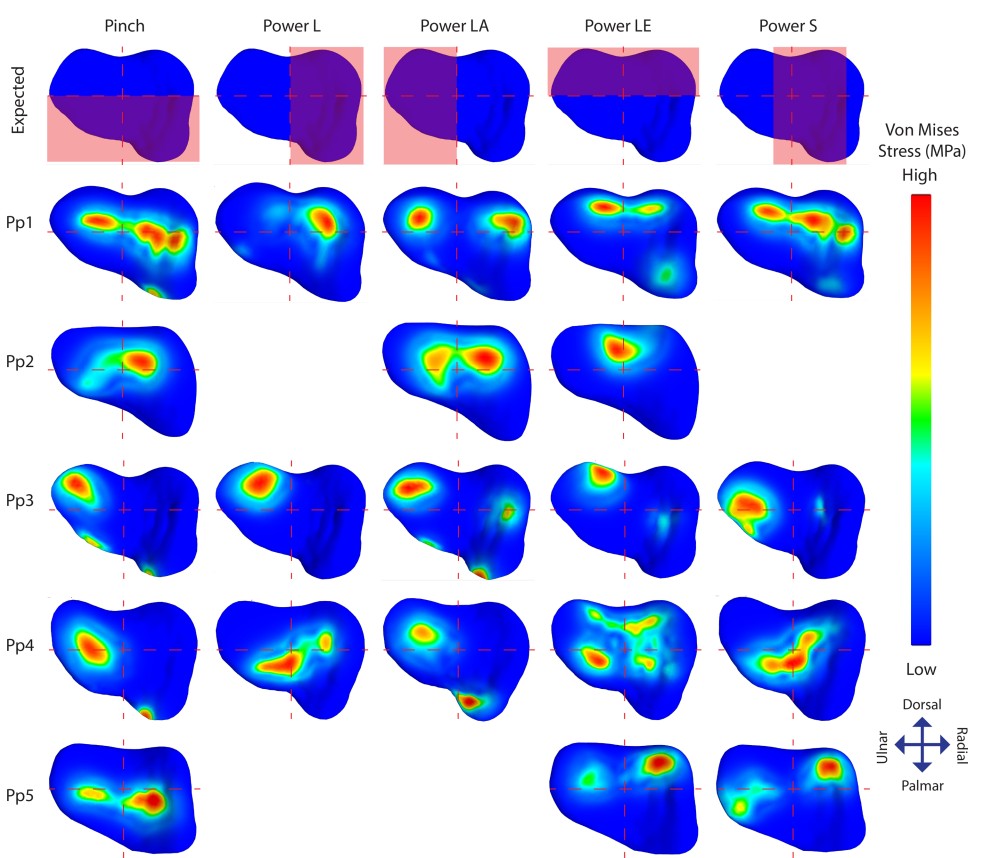

**Figure 5  TMC joint Von Mises stress distributions plotted on a distal view of the isolated trapezium distal articular surfaces.** The top row displays the quadrants where stress distribution is expected for each position in red. Empty slots represent unsuccessful scans or simulations.

unloaded state was accounted for in the final RMSE calculation by excluding the articular gap distance.

Distal trapezial articular surface Von Mises stresses were normalized and plotted for each individual model. The location of stress peaks and the overall stress distributions were compared to the expected patterns for each position.

## RESULTS

Out of 24 simulated models, 21 terminated successfully. Two grips of Pp2 (*powerL* and *powerS*), and one grip of Pp5 (*powerLA*) were unable to successfully terminate using our method due to issues discussed below. Average RMS errors for the successfully terminated models were 1.364 mm for *pinch*, and below 0.7 mm for the power grasp simulations (0.516 mm for *powerL*, 0.307 mm for *powerLA*, 0.514 mm for *powerLE*, and 0.667 mm for *powerS*).

Individual stress distribution plots (Fig. 5) show indications of the expected stress patterns in 18 out of the 21 simulations. During *pinch*, stress peaks are located centrally and palmarly when compared to the other positions in the same individual, cases of palmar

edge loading occur in 3/5 individuals. During *powerL*, radial stress distributions are present in two-thirds of the simulations. During *powerLA,* forces are directed to the ulnar side in the four successful simulations. The *powerLE* position displays high stress on the dorsal aspect of the articulation in all five simulations. During *powerS*, forces are diverted to the far radial side in one simulation, to the central-radial side in another simulation, and to a combination of both the central and far radial aspect of the articular surface in a third.

Stress distributions diverging from the expected patterns are present in 14 simulations; 11 show divergent patterns in addition to the expected patterns, and three show divergent patterns only (Fig. 5). Pp1 shows divergent patterns on the ulnar aspect of the articular facet during *pinch* and *powerS*, as well as on the radial aspect during *powerLA*. Pp2 shows high stress peaks radio-centrally during all three successful positions; *pinch*, *powerLA*, and *powerLE*, but during *powerLE* it is part of the expected location. Pp3 shows divergent stress peaks on the dorso-ulnar aspect of the articular facet during *pinch, powerL,* and *powerS,* these peaks are also present during the other two positions, but in those cases they are part of the expected pattern. In addition, Pp3 shows divergent indications of contact on the radial side during *powerLA*. Pp4 shows stress peaks located ventrally on the articular surface in all five positions, although these are only expected during *pinch*. Stress distributions in Pp5 are directed ulnarly in all three of the successful simulations, but this is only unexpected during *powerS*.

## DISCUSSION

The purpose of this study was to simulate functional grasps commonly used by bonobos and to investigate how stress distributions within TMC joint vary between these activities. We applied a computational modelling approach using parametric meshes to compare stress distributions across five individuals using five grasping tasks. The magnitude of the resultant force vectors were set to 300 N and the direction of the resultant force vectors were optimised in force-driven FE simulations for the 24 simulation cases. Of the 24 simulations, 21 simulations converged to an average RMSE of 0.673 mm ± 0.364 mm. We hypothesized that the simulated stress distributions for each position would correspond with the patterns expected from a saddle-shaped joint. However, we also expected there to be differences in stress patterns arising from individual variations in joint morphology.

Overall, the results appear to correspond well with our predictions for each position with 18 of the 21 succesfull simulations showing stress distributions largely matching the expected distributions for a universal type joint. Out of these, there are 11 cases where additional patterns of stress distribution can be recognized, while the remaining three simulations show patterns fully diverging from the expected stress distributions. In summary, the results show that grip type influences the stress distributions in the expected manner, but variations in joint morphology interfere with this pattern. Most likely, morphological variations cause the joint to function less like a universal joint. A perfect universal joint consists of a double hinge that allows for rotation in two planes only, while restricting axial rotation (*Cooney & Chao , 1977*). This strict functionality is by virtue of the shape of the joint and is lost when the morphology varies. In the current bonobo

sample, such variations are prevalent and consequently stress patterns appear to vary more between individuals than between grip types. A detailed interpretation of the individual differences is given below. Most of the deviations from the expected stress patterns can be explained by differences in joint morphology.

Our results give an indication of the influence of an individual's morphology on joint stress distribution (Fig. 5). The divergent patterns in Pp1 show high stress peaks on the dorsal aspect. Furthermore, there are indications of palmar contact during *powerLE* and *powerS*. When inspecting the TMC joint morphology of Pp1, these patterns coincide with the relatively protruding dorsal ridge and palmar beak of its MC1 (Fig. 6A), which come in contact with the dorsal and palmar parts of the articular surface of the trapezium upon loading. Pp2 shows stress peaks in the central-radial aspect of the articular surface in all three successful positions, while it was only expected in *powerLA*. This may be attributed to the dorsal ridge of its MC1 articular surface, which is more asymmetrical than in the other specimens. In Pp2, the most proximal point of the dorsal ridge is slightly offset to the radial side of facet. This causes contact with the trapezium, which symmetrical shape does not accommodate for the offset protrusion (Fig. 6B). During *pinch*, the asymmetrical dorsal ridge may initiate contact before the palmar beak. Furthermore, Pp2 is a sub-adult with a TMC joint that is not fully mature, resulting in a smoother articular periphery and in particular less pronounced dorsal and palmar ridges. This affects the shape of the TMC articular surfaces and appears to influence the interaction between the palmar beak and palmar aspect of the trapezium during *pinch* particularly, possibly causing it to not show the expected stress peaks in the palmar region during this grasp. Pp3 shows relatively high average stress peaks on the dorso-ulnar aspect of the articular facet in all positions. Upon visual assessment of this specimen it is clear that its joint morphology is relatively robust compared to the other specimens, *i.e.,* the morphological features of the trapezium and MC1 are more strongly developed. The dorso-ulnar periphery of the trapezium's distal facet protrudes heavily (Fig. 6C), interacting with the MC1 in every modelled scenario. All positions collide heavily on the ulnar aspect of the facet, causing high stress in that area for each position. Two of the three divergent patterns (that do not also show an expected pattern) are Pp3's *powerL* and *powerS*, which are both power grips with the thumb abducted. These two grips show the same divergent stress patterns. It appears that in these cases, the MC1 loads the protruding dorso-ulnar aspect of the trapezial facet's periphery instead of initiating contact on the radial side of the trapezial articular surface as expected for these grasps. This causes the protrusion to function as a bony stop, for which bone is not intended (*Simpson, Latimer & Lovejoy, 2018*). Such loading of the articular surface periphery may be the result of a self-perpetuating cycle where the high loads cause inflammation and pathological bone responses. Despite the absence of osteophytes, the protrusion may be an early sign of osteoarthritis (*Wilson, McWalter & Johnston, 2008*), although reports of hand osteoarthritis are rare in non-human primates. Also during *pinch*, high stresses are observed in the dorso-ulnar region, which can be explained by the protrusion on the articular periphery. Pp3's divergent stress patterns further show loading on the radial aspect of the trapezial articular facet during *powerLA*, *powerLE*, and *powerS*, as well as on the palmar edge during *powerLA*. This is likely due to the pronounced

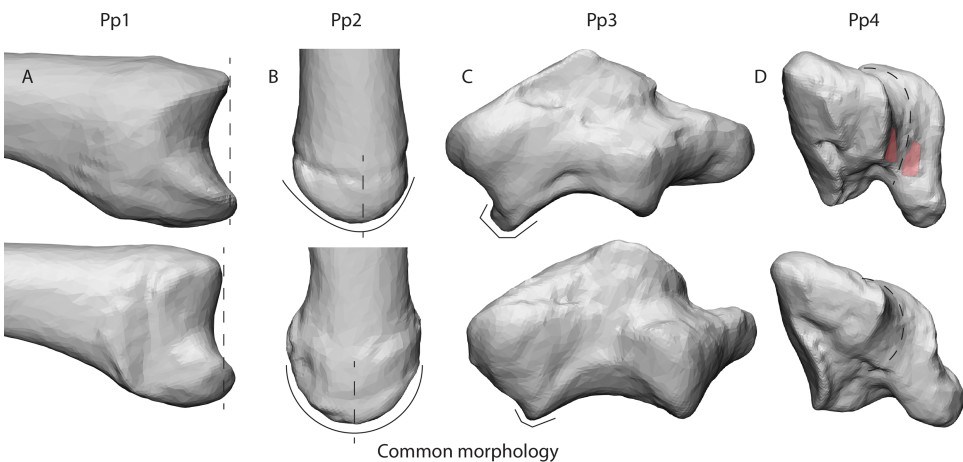

**Figure 6 Deviating morphologies in specimens Pp1-4 in the top row in comparison to more common morphologies of the remaining specimens in the bottom row.** (A) Radial view of the protruding dorsal ridge and volar beak of Pp1's MC1 base. (B) Dorsal view of the asymmetrical dorsal ridge of Pp2's MC1 base. (C) Dorsal view of Pp3's protruding dorso-ulnar periphery of the trapezial articular facet. (D) Disto-ulnar view of the Pp4's flattened palmar trapezial articular facet, as well as the palmar indentures highlighted in red.

dorsal ridge and palmar beak on the MC1. The divergent stress patterns in Pp4 indicate loading on the palmar aspect of the articular facet. The trapezial morphology shows that the palmar aspect of the articular surfaces flattens out relatively fast towards the palmar aspect (Fig. 6D). At the bottom, the surface shows light indications of indenture, which may capture the palmar beak during loaded positions (Fig. 6D). Pp5 shows divergent stress patterns on the ulnar side of the trapezial facet. This appears to be caused by a protrusion of the trapezial facet's ulnar periphery, as seen in Pp3 (Fig. 6C). The protrusion, however, is not as explicit as in the case of Pp3, which may explain the lower stress peaks in Pp5.

The bonobo is a primate species underrepresented in literature. Due to its remote and scarce population (*Prüfer et al., 2012*), academic focus on the bonobo has been limited in in comparison to its sister taxon; the chimpanzee (*Pan troglodytes*). Studies on the functional anatomy of the bonobo remain limited (*Miller, 1952*; *Vereecke et al., 2005*; *Payne et al., 2006a*; *Payne et al., 2006b*; *Myatt et al., 2012*; *Diogo, Molnar & Wood, 2017*), and most literature including *Pan* only make use of chimpanzees (*Lewis, 1977*; *Marzke et al., 2010*; *Rafferty, 1990*; *Rose, 1992*). Claims have been made that due to evolutionary stasis, the bonobo may be the most suitable model for the *Pan–Homo* last common ancestor (*Diogo et al., 2017*). Whether such a model exists in extant species is still debated, but it does indicate that the bonobo is a relevant piece of the evolutionary puzzle. Our previous research on the bonobo thumb revealed that their TMC joint is highly similar to that of humans, both in form and function (*van Leeuwen et al., 2018*; *van Leeuwen et al., 2019*). These similarities are further supported in the current study as the intra articular stress patterns encountered correspond well similar grasps tested in humans (*Schneider et al., 2017*). Chimpanzees also possess a similarly shaped TMC joint (*Lewis, 1977*; *Marzke et al., 2010*; *Rafferty, 1990*), likely with a similar capacity for mobility (*Rose, 1992*), and so

homologous joint loading patterns may be expected. However, caution should be taken when making such assumptions considering the behavioural differences between both species of *Pan* (*Hare, Wobber & Wrangham, 2012*; *Manson, Perry & Parish, 1997*). Bonobos are reportedly more arboreal (*Doran, 1993*), while chimpanzees appear to engage in tool use more often than bonobos (*Neufuss et al., 2017*). Yet these differences in reported locomotor and tool use behaviour may also be a reflection of the offset in study frequency between these taxa and need to be studied in detail. It would certainly be interesting to apply our FEA approach to a broader set of primate taxa, including not only the closely related chimpanzee but also primates displaying variations on the saddle-shaped TMC joint (*e.g.*, *Gorilla, Pongo, Papio, Macaca*), as well as those that display a completely different joint shape (*e.g.*, *Hylobatidae*) (*Rafferty, 1990*). To that end, this study provides a suitable FEA method to investigate joint interactions and stress patterns within these morphologies for future work.

We have developed an innovative modelling workflow for the estimation of intra-articular stress distributions in a joint, and demonstrated its performance in the bonobo TMC joint. We want to stress that, rather than aiming to provide accurate estimations of stress distributions in the bonobo TMC joint, our aim was to compare stress distribution patterns between grip types and between individuals. To increase the accuracy of the model, detailed information on non-human primate tissue properties and dimensions is needed. Since such information is still lacking, we make several assumptions when building our models. First of all, a simplified cartilage layer was included in our FE model as we were unable to obtain accurate cartilage dimensions for the studied specimens and had to rely on human-based studies. Furthermore, we modelled the cartilage layer as a layer with a constant thickness, while it has been shown that cartilage varies in thickness across the articular surface in the human TMC joint, be it only slightly (<10%) (*Dourthe et al., 2019*; *Koff et al., 2003*). In future modelling, we would recommend to include known dimensions of the cartilage layer as variations in cartilage thickness across the articular surface are likely to influence joint congruency and stress distributions. Inclusion of an accurate, non-uniform cartilage layer may affect the morphology of the TMC articular surfaces so that a more accurate representation of the stress distribution in the joint can be achieved. Our modelling approach has been designed with a modular cartilage layer to ensure that available information on TMC joint cartilage can be included in future studies. Secondly, the cartilage layer elastic properties were selected based on the previous human models (*Schneider et al., 2017*), and tuned within the range of human literature to the current values (Young's modulus = 18 MPa, Poisson's ratio = 0.45) (*Black & Hastings, 2013*). This to increase the robusticity of the modelling, and to ensure a higher success ratio throughout the large amount of models tested in various positions of the MC1. We expect that our choice for the parameters minimally affects the results of the study, as we compare differences in the distribution of stress, for which the exact properties have been shown to be less relevant (*Gil Espert, Nogué & Sánchez, 2015*).

While the sample size for this study was small due to inherent limitations in obtaining cadaver material from bonobos, a critically endangered species with a small captive population (*Fruth et al., 2016*), our approach allowed us to perform an in depth analysis

of form-function relation surpassing the level of detail of existing studies on non-human primates. We did, however, observe some sensitivity to joint model shape during certain wide grasps in our approach. Due to this sensitivity, not all models were able to terminate successfully and we had to eliminate three simulations during our analysis. Pp5's *powerLA* position was excluded due to severe cases of edge loading in the later stages of the simulation. Edge loading occurs as a result of the lofting approach used to simulate the TMC cartilage layer, where the edge of the cartilage is artificially high compared to the surrounding bone. This mainly presents an issue during simulations of extreme MC1 excursions where the only remaining contact with the trapezium is maintained directly onto this artificial cartilage edge. Such extreme loading on the edge cause the peripheral elements of the mesh to collapse, and the simulation to fail. We furthermore had to exclude two grasp types of the sub-adult specimen Pp2. The relatively smooth morphology of the articulating bones presented difficulties in simulating grasps with an abducted thumb (*powerL* and *powerS)* and these positions could not be successfully terminated using our approach. Despite the low sample size, the open source FE method presented here is widely applicable for the study of stress distributions in joints, regardless of species or joint system.

This study yields biomechanical insight in the bonobo thumb, moreover it provides subchondral stress distributions of the TMC joint which will be included in future research on joint morphology in bonobos. The exact locations and magnitudes of these forces will be applied to investigate if the orientation of the trabecular microarchitecture of the bonobo trapezium correlates with the stress distributions estimated here. This to investigate how the bonobo thumb morphology is adapted to different types of manual activities. Moreover, it will provide in depth information that is directly relevant for the interpretation of fossil remains.

## CONCLUSIONS

The present study applies a parametric FE approach to simulate stress distributions in the TMC joint of bonobos with the aim of investigating how stress distribution differs during five grasps commonly employed in captive populations. We find that the simulated stress patterns correspond well with the expected patterns during the modelled grasps, and the divergent signals can be attributed to the individual joint morphology. We conclude that interindividual variation in joint morphology has a strong influence on intra-articular stress distributions and should always be considered when modelling joint interactions. The results demonstrate that this modelling workflow is suitable to estimate TMC joint stress distributions and can be a valuable tool for the study of primate joint biomechanics in future work.

## ACKNOWLEDGEMENTS

The authors are indebted to Jeroen Stevens for his initiative taken in the collection of bonobo specimens and his courtesy in collaborating with our lab to make this study possible. We are thankful to Maarten Vanneste, Ted Yeung, Nynke Rooks, and Mousa Kazemi for their expertise and input on modelling and programming. We extend our

gratitude to Olivier Vanovermeire and Henk Lacaeyse at the Medical Imaging Department, AZ Groeninge (Kortrijk, Belgium) for providing us with the means and assistance for the imaging protocol. Furthermore, we thank Marie Vanhoof and Kristiaan D'Aout for their courtesy in supplying us with primate images. The authors thank Johanna Neufuss, Tracy Kivell, and one anonymous reviewer for their valuable input which greatly helped with the improvement of this manuscript.

### Funding

This work was supported by the KU Leuven (no. RQ1-D1114-C14/16/082) and the Research Foundation Flanders (no. V415519N). The funders had no role in study design, data collection and analysis, decision to publish, or preparation of the manuscript.

### Grant Disclosures

The following grant information was disclosed by the authors:
KU Leuven: RQ1-D1114-C14/16/082.
Research Foundation Flanders: V415519N.

### Competing Interests

The authors declare there are no competing interests.

### Author Contributions

- Timo van Leeuwen conceived and designed the experiments, performed the experiments, analyzed the data, prepared figures and/or tables, authored or reviewed drafts of the paper, and approved the final draft.
- G. Harry van Lenthe and Evie E. Vereecke conceived and designed the experiments, authored or reviewed drafts of the paper, and approved the final draft.
- Marco T. Schneider conceived and designed the experiments, performed the experiments, analyzed the data, authored or reviewed drafts of the paper, and approved the final draft.

### Animal Ethics

The following information was supplied relating to ethical approvals (i.e., approving body and any reference numbers):

Not applicable, only cadaveric non-human primate specimens were used. No formal approval of the animal ethics committee is required.

### Field Study Permissions

The following information was supplied relating to field study approvals (i.e., approving body and any reference numbers):

Wilhelma Zoo, Stuttgart, Germany

Royal Zoological Society Antwerp, Belgium

Zoo Frankfurt, Germany

La Vallée des Singes, Le Gureau, France

## Data Availability

The imaging data is available in Raw and STL format at MorphoSource:

- Pp1 - Pinch, DOI: 10.17602/M2/M380974
- Pp1 - PowerL, DOI: 10.17602/M2/M380980
- Pp1 - PowerLA, DOI: 10.17602/M2/M380986
- Pp1 - PowerLE, DOI: 10.17602/M2/M380992
- Pp1 - PowerS, DOI: 10.17602/M2/M380998
- Pp2 - Pinch, DOI: 10.17602/M2/M381004
- Pp2 - PowerL, DOI: 10.17602/M2/M381011
- Pp2 - PowerLA, DOI: 10.17602/M2/M381019
- Pp2 - PowerLE, DOI: 10.17602/M2/M381027
- Pp2 - PowerS, DOI: 10.17602/M2/M381033
- Pp3 - Pinch, DOI: 10.17602/M2/M381039
- Pp3 - PowerL, DOI: 10.17602/M2/M381046
- Pp3 - PowerLA, DOI: 10.17602/M2/M381052
- Pp3 - PowerLE, DOI: 10.17602/M2/M381058
- Pp3 - PowerS, DOI: 10.17602/M2/M381064
- Pp4 - Pinch, DOI: 10.17602/M2/M381077
- Pp4 - PowerL, DOI: 10.17602/M2/M381093
- Pp4 - PowerLE, DOI: 10.17602/M2/M381118
- Pp4 - PowerS, DOI: 10.17602/M2/M381136
- Pp5 - Pinch, DOI: 10.17602/M2/M381145
- Pp5 - PowerLA, DOI: 10.17602/M2/M381152
- Pp5 - PowerLE, DOI: 10.17602/M2/M381159
- Pp5 - PowerS, DOI: 10.17602/M2/M381181

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
