# Peer review of "Stress distribution in the bonobo (Pan paniscus) trapeziometacarpal joint during grasping"

_PeerJ, doi:10.7717/peerj.12068_

## Round 0.1 · original submission · Major Revisions

Three reviewers have perused the study and two have provided quite positive reviews while one is more critical but still confident that issues with the study should be resolvable. Thus all 3 reviewers agree the study has strong potential for publication. Overall the revisions required are moderate in nature; from improving the literature review to clarifying and perhaps adjusting the FEA methods/analysis and figures. Please ensure you address all points individually in a Response. Some re-review will be necessary. Thank you!

·

Basic reporting

No comment.

Experimental design

No comment.

Validity of the findings

No comment.

Additional comments

Overall, this is an original study that provides needed information on the form-function relationship between the TMC joint morphology and hand grip use in bonobos; a primate species that is very under-represented in anatomical and biomechanical studies on great ape hand use. The manuscript is written in a clear and accessible style, enabling relatively easy appreciation of what can sometimes be an overly-technical topic. The small sample size is quite understandable, given the difficulties in obtaining cadaveric specimens of captive bonobos. Despite this limitation, the methods of data collection and analysis are appropriate, increasing the power of this study. The figures illustrate the text well and present the data clearly. My only substantive comments are quite minor.

Reviewer 2 ·

Basic reporting

The manuscript submitted by van Leeuwen and colleagues presents the results of an interesting study that analyses how the trapeziometacarpal (TMC) joint morphology influences stress distributions of the joint in five grasping types used by bonobos. In order to do so, they simulate TMC joint articular force distributions using FEA to evaluate stress patterns associated with each one of these different grasp types, using an innovative modelling workflow. The authors state that their models show a high agreement between simulated and expected stress patterns, however, they also found ‘deviating patterns’ that they attribute to morphological differences. I consider that the manuscript is appropriate for the journal and of significant interest, but I have also identified a number of issues that need to be addressed prior to an eventual publication. In particular, the ‘Introduction’ section does not adequately review the relevant literature on the morphology and function of the TMC joint in apes; some details about the methods are not fully developed; and the ‘Discussion’ does not contextualise the findings with respect to the relevant literature. Below, I provide a number of ways in which the manuscript can be improved.

Main comments:

Introduction: For decades, an extensively researched topic relates to the evolution of the thumb in apes and humans, having been addressed from several different perspectives. As such, there is a vast corpus of studies that is certainly relevant to this manuscript. Even though I do understand the value of brevity, there are a number of omissions that hinder the purpose of contextualising this study within the broader literature of the area. The introduction of this manuscript does indeed provide some classic and relevant citations (e.g., Napier, 1955; Cooney and Chao, 1977; Napier, 1956, among others), which although important and valuable, are not the only sources of information available nowadays. The radial carpometacarpal (CM) joints include the complex articulations between the scaphoid, trapezium, trapezoid and the first and second metacarpals, and within the CM the trapeziometacarpal (TMC) has received the most attention, both in what regards primate morphology, as well as in clinical studies and as such there are a plethora of studies that might be relevant to your introduction (to name a few additional examples Haines 1944; Tuttle 1969; Kuczynski 1974; Lewis 1977, 1989; Rafferty 1990; Rose 1992, among many others).

In its current form, the introduction and discussion both feel a bit rushed, which is evidenced by a poor literature review. Hence, I consider that the authors should try to improve their introduction by providing additional references and information regarding the importance of the TMC and why it is relevant to study its form and function. This would help you better organise your goals, as well as strengthening the relevance of your study from a biological perspective. In summary, the introduction needs a higher level of detail. I think you should improve the literature review so as to provide more justification and better contextualisation for your study. In addition, you should try to expand on the knowledge gap that is being filled, and how this is relevant from a biological and evolutionary point of view.

Line 54: The authors mention that bonobos, like humans, possess a saddle-shaped TMC joint, which is true. However, it has been also argued that a saddle-shaped joint TMC may be the primitive condition for mammals (Lewis 1977). In fact, across cercopithecoids, the TMC joint is usually saddle-shaped with modest variation in bony morphology (Rafferty 1990). Even in colobines that have a tiny thumb, the TMC joint is saddle-shaped. This means that a saddle-shaped TMC and its similarity to humans is not a good enough justification. What is particularly unique among all great apes and is different with respect to non-hominoid primates, is that their well-developed saddle-shaped TMC joint allows them to have a much greater abduction-adduction mobility (Rose 1992). You could use this information to better justify the importance and relevance of your study.

Line 52: Please explain why only bonobos were studied, and why you did not include any common chimpanzees (Pan troglodytes) to compare results between these sister taxa. As you correctly point out in your abstract, Pan paniscus is a species under-represented in morpho-functional studies despite its endangered conservation status and close phylogenetic relation to modern humans. Please explicitly mention the reasons underlying your decision to exclusively focus on bonobos. It would be also relevant to mention the locomotor and tool use differences between bonobos and chimpanzees.

Lines 64-66: You mention that your previous research on the kinematics of the TMC joint showed that the bonobo thumb is highly mobile (van Leeuwen et al., 2019). Please briefly expand on this; highly compared to whom? Why it appears that some structural restraints from soft- and/or hard tissue appear to restrict the bonobo’s thumb? etc.

Lines 72: You studied five different grasps (that you named pinch, powerL, powerLE, powerLA, and powerS). Please briefly explain why you decided to focus on these five different grasps in particular.

Lines 72-89: This may be a matter of taste, but this paragraph could be transformed into explicit hypotheses and organised in a clearer way (e.g., a table).

Experimental design

There are some aspects of the methods that require additional information.

Material and methods:

Specimens: The fact that your sample mixed captive and wild-born individuals may be somehow problematic. A captive animal (which was also raised in captivity) would not necessarily show the locomotor repertoire of individuals that lived in the wild. You should consider assessing for differences across these different origins, as perhaps this factor may explain some of the differences you observed in your results. In any case, a short sentence expressly mentioning this possible issue should be added, as it has been shown that there are differences in the articular surfaces of captive and wild chimpanzees (see for e.g., Lewton, 2017).

• Lewton, K. L. (2017). The effects of captive versus wild rearing environments on long bone articular surfaces in common chimpanzees (Pan troglodytes). PeerJ, 5. https://doi.org/10.7717/peerj.3668

It has been shown that bonobos exhibit hand-preferences (e.g., right-hand bias) when performing certain tasks (Neufuss et al., 2017). Can hand-preference and the fact that you mixed left and right arms affect your results in any way?

• Neufuss, J., Humle, T., Cremaschi, A., & Kivell, T. L. (2017). Nut-cracking behaviour in wild-born, rehabilitated bonobos (Pan paniscus): A comprehensive study of hand-preference, hand grips and efficiency. American Journal of Primatology, 79(2), 1–16. https://doi.org/10.1002/ajp.22589

In addition, please explicitly mention that no articular pathologies were observed at the TMC because otherwise those individuals should be removed from the sample.

Methods:

Lines 158-160: You mention that as there is no available cartilage information for bonobos, so you arbitrarily decided to use a thickness value of 0.5 mm, which is somewhat lower than the values reported for the human TMC joint cartilage (Dourthe et al., 2018) in order to account for the relatively smaller bone size. Although a reasonable assumption, this should be tested in a better way, particularly if we consider that among the ‘Critical considerations’ at the end of your manuscript, you mention the fact that you modelled cartilage as a layer with a constant thickness using an arbitrarily defined value. You should run a sensitivity test varying the thickness of this layer (and ideally model cartilage not using a constant thickness), to quantify the effect of the error introduced by this factor.

Line 172: Please provide a reference for your material properties. In addition, what values did you use for bone? Did you model the underlying bone as solid cortical bone? Please provide additional information.

Line 195: Why did you only use the RMSE? This needs to be justified. There is some discussion regarding which metric error to use in these cases. It would be better if you could compare the performance of a sub-set of models using at least one different error metric (e.g. mean absolute error [MAE] or another one), as there might be important differences, particularly when considering your small sample size.

How did you account for size differences in your results? When comparing different models exhibiting size differences it is necessary to adjust the models to a comparable size. There are several ways of answering the question of how performance can be compared so that the effects of size and shape are disentangled (see for e.g. Dumont et al. 2009). Among the most common options are: (1) modify the size of all models so that all have the same size and then apply the same force, or (2) keep the differences in size, and then apply an appropriate force that generates the same effect as carrying out the procedure described in point (1). Please provide additional information.

• Dumont, E. R., Grosse, I. R., & Slater, G. J. (2009). Requirements for comparing the performance of finite element models of biological structures. Journal of Theoretical Biology, 256(1), 96–103. https://doi.org/10.1016/j.jtbi.2008.08.017

Validity of the findings

Results:

How did you compare the results from your FE models? FEA models and their respective results have been usually compared in a mostly qualitative way in functional-morphology, by visually comparing stress or strain distributions (as you do in your Fig.5 where you do not provide the actual range of your MPa values in your von Mises rainbow scale). This qualitative assessment of FEA results has been usually done by looking at contour plots, and it is a straightforward option when the number of models and loading scenarios is small, as well as when the obtained results are distinctive enough (which are two conditions that I do not think necessarily apply to your results). For these reasons, you should explore more quantitative ways to compare your obtained results (see for e.g. Marcé-Nogué 2020 for some options) or other additional alternatives such as techniques that have been developed to conduct statistical inference on smooth biomechanical field data of any physical dimensionality (e.g. statistical parametric mapping; Pataky 2010). Using some of these options would allow you to quantitively evaluate your results, hence enriching your manuscript.

• Marcé-Nogué, J. (2020). Mandibular biomechanics as a key factor to understand diet in mammals. https://www.researchgate.net/profile/Jordi-Marce-Nogue/publication/348266506_Mandibular_biomechanics_as_a_key_factor_to_understand_diet_in_mammals/links/5ff5995a92851c13feefff87/Mandibular-biomechanics-as-a-key-factor-to-understand-diet-in-mammals.pdf
• Pataky, T. C. (2010). Generalized n-dimensional biomechanical field analysis using statistical parametric mapping. Journal of Biomechanics, 43(10), 1976–1982. https://doi.org/10.1016/j.jbiomech.2010.03.008


Discussion: Although the discussion properly summarises your findings, there are no references in the whole section. Please contextualise your results with respect to the relevant literature. In particular, since you mention the evolutionary importance of understanding the form and function relation of the thumb in your introduction, you should try to contextualise your results from a biological and evolutionary point of view. This will certainly improve your manuscript, highlight its relevance and would probably widen your readership.

Additional comments

Additional comments:

• I would recommend mentioning in the abstract that you used finite element analysis (FEA) to carry out your simulations rather than simply saying that you generated finite element models, as a wider readership interested in functional morphology may be more familiar with the FEA concept/acronym.

• Please include the versions of all the software used in your study (e.g. lines 145, 151, etc.)

• Although I am aware that the rainbow colour plots have a long history in FEA, using rainbow colour plots can be visually misleading (as it can distort perceptions of data and alter meaning by creating false boundaries between values), as well as being problematic for colour-blind readers. Hence, I suggest altering the colour schemes used in your FEA plots (Fig. 5). See for additional information:


https://www.climate-lab-book.ac.uk/2014/end-of-the-rainbow/
https://root.cern.ch/rainbow-color-map
https://eagereyes.org/basics/rainbow-color-map

·

Basic reporting

This study uses finite element modelling to estimate the areas of stress experienced by the bonobo TMC joint during 5 different grip postures. Overall, this is a robust study including 5 bonobo cadavers; an absolutely small sample size but relatively large and impressive when considering how difficult it is to access great ape cadaveric specimens. The manuscript is very well written, the structure is well organised and clear, and the figures are of excellent quality and relevant to the paper. Overall, I think the study is a welcome contribution to the literature on great ape (including human) thumb and hand function. I have several comments and questions below, most of which are relatively minor or ask for clarifications about what has been done. However, I have no major concerns with the analysis or interpretation.

The introduction is a little light on citations, particularly of early/primary research, and I have made several suggestions below that could be added, in addition to addition minor edits throughout the manuscript.

line 41: This general statement should have primary citations, rather than Samuel et al. (2018). I would suggest early papers by Napier and most papers by Mary Marzke (many of these are cited elsewhere in the manuscript already).
57: would be good to clarify that this is mainly tool use in captivity, not the wild. You could also include Neufuss et al. (2017) on bonobo nut-cracking in wild-born, sanctuary bonobos (which is incorrectly cited elsewhere in the paper)
62: following on above, Neufuss et al. (2017) is not appropriate to cite in reference to bonobo thumb use during locomotion since this paper is about tool use. Neufuss et al. (2017) in AJPA on hand/thumb use during vertical climbing in chimpanzees (and mountain gorillas) could be cited instead (perhaps this was the paper the authors meant to cite?). Also, the primary work of Tuttle or Inouye (she spoke more directly to bonobos) on knuckle-walking hand postures should also be cited/acknowledged here.
65: change ‘restrains’ to ‘constraints’
76: not clear what is meant by “diverted to the radial, but central aspect of the articular facet”. Is radial or central? Or radial-central?
Figure 2 is very helpful for understanding the different grips. I suggest moving the ‘pinch grip’ image ‘B’ to bottom R in this figure so it is consistent with the order in which the grips are described within the text.
104: change “five functional positions; three types of large diameter power grasps, a small diameter power grasp and a key pinch precision grasp” simply to “five functional grip positions.” and delete the rest since these have already been described in detail in the Introduction.
141: I think there is a word missing here: “four), one per grasp.” Perhaps it is just ‘and one per grasp’ but it’s not clear.
144: Move ‘triangular’ to in front of ‘meshes’ here. Otherwise it implies the trapezium and Mc1 are triangular.
Figure 3. It is not clear to me what is meant to be represented by the image directly under ‘D’. This looks like a rabbit in ramakin and chop sticks to me so have no idea what it is included in this figure.
Figure 5 is excellent – very reader friendly and easy to follow!
246: change to ‘individual’ or ‘intraspecific’ variations in morphology.
252: change to “morphology interfere with this…”
268: change ‘Here’ to ‘In this individual,’

Experimental design

I am not expert in FE modelling so cannot critically evaluate all of the methodological details. However, I understand the basics of this method and the basics of what has been done in this study in particular, in part because the methods are described quite clearly. However, there are some aspects I do not fully understand and have queried below. Some may stem from my own ignorance, but may still be worth clarifying in the manuscript for non-specialist readers.

1. One of the sample specimens is 8 yrs old (while the remaining 4 are adult). Please provide some information in the ‘Specimens’ section about morphological (bone ossification and soft tissue) and behavioural development at 8 years of age in bonobos and why you think it is suitable to include a subadult in the sample. Do you expect this individual to be an outlier in anyway? This is addressed somewhat in the Critical Considerations section, but should also be addressed here.

2. What are the ‘bone landmarks’ referred to in line 147? Are these particular 3D landmarks that were identified on each bone and, if so, what are they? Or is this meant to refer to generally fitting to two articular surfaces together in a way that makes functional sense given that the a saddle-joint articular does not allow for a lot of ‘wiggle room’ in how you can put the two bones together?

3. I have questions about the following statement: “We performed a 2D static force analysis, based on Cooney and Chao (1977), to estimate the magnitude of the resultant force vector applied on the thumb tip. In the static force analysis, the applied force was estimated at 50N, 1/6 of the average bonobo body weight (30kg)…” Where does the 1/6 of body weight come from? Is this because Cooney & Chao estimate a particular load at the TMC joint humans and this load in 1/6 of the average human body mass? Clarity around 1/6 (vs. for example, ¼ or 1/8) in particular has been chosen is needed.

4. It would be helpful to add to the end of the methods what criterion(ia) are used to determine a ‘successful’ termination. The authors state that “force vector to yield a loaded terminating position that closely resembles the unloaded” but at the start of the results there are measures provided in mm for the successfully terminated models. Was there a range or threshold of ‘acceptable’ RMS errors you were looking for and, if so, what was it?

Validity of the findings

General summary of the results accurately reflects the findings. The authors provide a clear and honest assessment of what they found, and an excellent discussion of results that did not fit expectations.

1. I would like to read greater clarity about this statement: “Most likely, morphological variations cause the joint to function less like a the universal joint.” For a non-specialist reader, it would be helpful to clarify what a universal joint is and generally how individual morphological variation might affect this. I know the next paragraph goes into detail about this morphological variation, but this statement is a bit unclear in this initial paragraph.

2. line 271: “During pinch, this aspect may initiate contact before the volar beak. Furthermore, Pp2 is a sub-adult with relatively under-developed joints. The overall morphological landmarks are comparatively smooth and nuanced. This affects the shape of the TMC articular surfaces and appears to influence the interaction between the volar beak and palmar aspect of the trapezium during pinch particularly, plausible causing it to not show the expected stress peaks in the palmar region during this grasp.” Just a few points of clarity needed here. In the first sentence, replace ‘this’ with “asymmetrical dorsal ridge” so it clear what aspect of morphological variation is being discussed. What exactly does ‘underdeveloped joints’ mean and ‘morphological landmarks’? Are the landmarks the same as those referred to in the methods? What does ‘nuanced’ mean in relation to the morphology exactly? In the last sentence, change ‘plausible’ to ‘possibly’. Looking at Figure 6, it is clear the epiphyseal fusion is not complete, which I think should be acknowledge somewhere in the manuscript (see comment in Methods above).

3. Related to above, Figure 6 is very helpful in visualising the morphological variation. However, I do have a few questions about this. In Fig 6a, it looks to me like the ‘common’ morphology has a more protruding volar beak than Pp1. Or perhaps this is better described as deeper dorsovolar curvature?

4. line 285: in reference to statement “Mc1 gets caught on the protruding….”. It is clear what the authors mean here, but I think it worth reflecting on descriptions of joint function by Lovejoy and colleagues. Simpson et al. (2018, Anatomical Record) discuss the function of synovial joints (see their pg. 499) in relation to the dorsal ridge in metacarpals and how the ridge does not ‘stop’ the proximal phalanx movement. I think the same rationale applies here. I don’t necessarily disagree with the author’s interpretations about the TMC joint but think it is worth (re-)reading Simpson et al. and related work cited in their paper, and perhaps just using more careful language about how morphological variation in the joint surface might affect where the joint stress occurs.

5. The Critical Considerations section is well written and excellent addition to the paper.
If the results for some positions for Pp5 had to be excluded and it was not possible to put this cadaveric specimen into the powerL position due to lack of mobility, how might this affect the results of the positions you did include? Did this specimen show signs of osteoarthritis? Or do you feel this reduced mobility was due to soft tissue changes?

Additional comments

Minor additional comments:
1. Abstract: change “The thumb is a crucial structure in primate evolution due to its role in grasping AND THE TRAPEZIOMETACARPAL (TMC) JOINT IS CRITICAL TO ITS FUNCTION” or something similar to avoid using ‘role’ twice.

---

## Round 0.2 · accepted · Accept

Congratulations-- you have nicely convinced the reviewer and the outcome is very positive. Well done!

Reviewer 2 ·

Basic reporting

no comment

Experimental design

no comment

Validity of the findings

no comment

Additional comments

I congratulate the efforts made by the authors to incorporate (or refute) the comments made by the different reviewers. I am satisfied with the changes they have made as well as with their explanations as to why they decided to follow (or not) the recommendations I provided during the first revision round. I think that the current version of the manuscript has certainly been improved. In particular, the change of emphasis made by the authors is a correct decision, as it helps to further strength the novel methodological aspects of this work.